# Fully Ordained Nuns in Fourteenth-to-Seventeenth Tibetan Hagiographical Narratives

**Fan Wu**

Department of Religious Studies, University of Virginia, Charlottesville, VA 22903, USA; fw9bs@virginia.edu

**Abstract:** Many contemporary efforts have been put to (re-)establish the order of fully ordained nuns in Tibetan Buddhism. Those who are in favor of such practice often refer to premodern Tibetan hagiographies to claim the existence of indigenous fully ordained nuns in the past. A series of female practitioners, indeed, appear as fully ordained nuns in such narratives dating from approximately the fourteenth century to the seventeenth century. Their monastic identities as such, however, are contested by Tibetan Buddhist masters because the methods of their ordinations, seemingly conferred by the male *saṃgha* alone, do not strictly follow the Mūlasarvāstivāda *Vinaya* tradition, which is observed by the Tibetan Buddhists. In an effort to investigate as to how these female practitioners were fully ordained and the purposes of composing such narratives about their ordinations, this article revisits relevant hagiographies with particular reference to *The biography of Chokyi Dronma, the Third incarnation of the Wisdom Ḍākinī Sonam Peldren* (*Ye shes mkha' 'gro bsod nams dpal 'dren gyi sku skye gsum pa rje btsun ma chos kyi sgron ma'i rnam thar*) and a detailed exposition of *The biography of Shākya Chokden* (*Shākya mchog ldan gyi rnam thar zhib mo rnam par 'byed pa*). It suggests that depicting these personas as fully ordained nuns serves the purpose of highlighting the hagiography subjects' outstanding spiritual performance, while the recognition of monastic identity as such may not go beyond the context of these writings.

**Keywords:** Tibetan hagiographies; Tibetan women; full female ordination



## 1. Introduction

Fully ordained nuns (Skt. *bhikṣuṇī*, Tib. *dge slong ma*) serve as an essential organizational constituent to the Buddhist institution from the Buddha's time onwards. The order of fully ordained nuns, however, had not been firmly established when Buddhism was introduced to Tibet. The vast majority of nuns in the Tibetan Buddhist community, therefore, have not historically had access to full ordination, and they could merely be semi-ordained (Tib. *dge tshul ma*) as their possible highest position, while monks have had uninterrupted access to the higher ordination. This phenomenon highlights that the Tibetan nuns receive insufficient recognition as proper members of the Buddhist monastic circles and this restriction to full ordination might hinder them from becoming qualified Buddhist masters (see also Schneider and Price-Wallace in this volume).

This issue has gained special attention from both the Tibetan Buddhist community as well as academia[1]. In an effort to explore possible solutions to (re-)introduce[2] the full female ordination, prior scholarship has done preliminary research on a series of hagiographies, dating from approximately the fourteenth century to the seventeenth century[3]. Such writings indicate that fully ordained nuns were active sometime from the twelfth century to the sixteenth century. Martin (2005) has conducted a study on Machik Ongjo (Ma gcig ong jo; circa twelfth century) and Konchog Tsomo (Dkon mchog gtso mo; u.d.) and asserts that the selected figures received full ordination from their male masters Khyungtsang Repa (Khyung tshang ras pa; 1115–1176) and Je Mikyod Zhab (Rje Mi skyod zhabs; u.d.), respectively. Diemberger (2007) has translated a hagiography devoted to Chokyi Dronma (Chos kyi sgron ma; 1422–1455) and suggests that she was not only fully ordained by her

first master, Bodong Chogle Namgyal (Bo dong Phyogs las rnam rgyal; 1376–1451), but also regarded as a tantric consort (Tib. *phyag rgya ma*) for both Chogle Namgyal and Thangtong Gyalpo (Thang stong rgyal po; 1385–1464). Price-Wallace (2015), based on contemporary Tibetan scholar Rinchen Ngodrup's work *A New Explanation of the Hundred Controversial Issues Regarding Fully Ordained Nuns: All Wish- fulfilling Treasure, the Beryl Collection* (*Dge slong ma'i gnad brgya pa sngon med legs par bshad pa'i gter dgos 'dod kun 'byung bad'urya'i phung po zhes bya ba bzhugs so*), enumerates a handful of fully ordained nuns together with their bibliographical data. They, however, mainly focus on these female personas as being depicted as fully ordained nuns and pay limited attention to the relevant questions surrounding their ordinations. To be specific, how were these nuns fully ordained? Why are they not widely recognized by Tibetan Buddhists? What are the purposes of composing such narratives? This article will investigate these points and attempts to offer insight into how Tibetan Buddhists historically viewed the issue of full female ordination.

## 2. Buddhist Hagiographies in Premodern Tibetan Context

Hagiographies (Skt. *vimokṣa*, Tib. *rnam par thar pa*) in the Tibetan Buddhist context highlight the doctrinal, contemplative, and social significance of distinguished practitioners by weaving his/her relevant information, either factual or fabricated, into life writings. They are also recognized by scholars as valuable sources for studying Tibetan political history and written traditions of religious communities (Sernesi 2015, pp. 734–35). One of the main doctrinal purposes of hagiographies lies in edification. To be specific, they are supposed to inspire later practitioners to follow virtuous models (Sernesi 2015).

The mainstream of this kind of Buddhist narrative centers on prominent male practitioners that have been discussed at length by previous scholars. For instance, Quintman (2010, 2014) has conducted fundamental research on multiple versions of the Tibetan cultural hero Milarepa's (Mi la ras pa; 1052–1135) life stories and compiles a comprehensive collection of prior versions of Milarepa. Caumanns (2010, 2015) has carried out studies on more comprehensive and detailed hagiographies devoted to an authoritative master of later dates, Shākya Chokden (Shākya mchog ldan; 1428–1507). On the contrary, distinguished female practitioners (i.e., Yeshe Tsogyal [Ye shes mtsho rgyal], Machig Labdron [Ma gcig lab sgron], Chokyi Dronma) have received relatively limited scholarly attention (Liang 2020; Allione 2000; Diemberger 2007). There are six texts which detail the full ordination as nuns of individuals or even entire groups—the references are as follows:

*The Short Biography of Machik Ongjo (Ma gcig ong jo'i rnam thar zur tsam*; hereafter MRZ): MRZ is collated in the existing Biography of Chakrasamvara Hearing Lineage (Bde mchog snyan brgyud kyi rnam thar skor), a collection of hagiographies of masters in the lineage of the oral transmission of Rechungpa (Ras chung snyan brgyud). Thanks to the colophon, it can be attributed to Rinchen Gyatso (Rin chen rgya mtsho; u.d.) and mainly circulated within the circle of the Marpa Kagyu (Mar pa Bka' brgyud) school, a subdivision of the Kagyu tradition. MRZ highlights Machik Ongjo, a female practitioner as well as a holder of the oral transmission of Rechungpa, and alleges that she was active in the twelfth century. The text details her outstanding spiritual performance and a full ordination conferred by her master, Khyungtsang Repa, who inherited the oral teaching from Rechungpa Dorje Drakpa (Ras chung pa Rdo rje grags pa; 1085–1161). Among other narrative constituents, MRZ distinctively emphasizes the inevitability of Khyungtsang Repa's prophecy about her ordination.

*Addendum to the Sakya Genealogy Marvelous Storehouse* (*Sa skya'i gdung rabs ngo mtshar bang mdzod*; hereafter SGM): This text, a genealogy of the Sakya school that features outstanding Sakya masters' lives and their achievements, was compiled by Ngawang Kunga Sonam (Ngag dbang kun dga' bsod nams; 1597–1659), the twenty-seventh throne holder of the Sakya School. Relevant parts of it praises the fifth of the five Sakya Patriarchs, Drogon Chogyal Phagpa's ('Gro mgon chos rgyal' phags pa; 1235–1280) *Vinaya* practice by asserting that once he conferred ordination to over a hundred monastics, including fully

ordained nuns. This text is originally preserved in Tibetan and its Chinese translation is available (Chen 2005).

*The Biography of the Great Khenpo, the Learned one Rigpa Senge: The Ocean of Precious Merits (Mkhan chen bka' bzhi pa chen po rig pa'i seng ge'i rnam thar pa yon tan rin po che'i rgya mtsho*; hereafter MRR): this text devoted to a revered scholar of Minyak Rigpa Senge[4] (Rig pa seng ge; 1287–1375) was composed by his close disciple Senge Zangpo (Seng ge bzang po; u.d.). Thanks to the colophon, it is considered to be an important historical source concerning the Minyak (Mi nyag) region of the Kham area. This hagiography suggests that a fully ordained nun and several monks dwelled in Rabgang (Rab sgang) monastery in Minyak region, and they nurtured an assembly of fully ordained nuns there. It also records that fully ordained nuns under his supervision practiced meditation and studied Vinaya and other Buddhist texts. Schneider (2012) has cited this text in her article on the issue of fully ordained nuns in Tibetan Buddhism.

*The Biography of Chokyi Dronma: The Third Incarnation of the Wisdom Ḍākinī Sonam Peldren (Ye shes mkha' 'gro bsod nams dpal 'dren gyi sku skye gsum pa rje btsun ma chos kyi sgron ma'i rnam thar*; hereafter CR): This incomplete hagiography is devoted to Chokyi Dronma, and the author and the dating of the text are unknown. It displays her secular and religious life; it also narrates how Chokyi Dronma was fully ordained and implies that she might have been a tantric consort of her master (Diemberger 2007).

*A Detailed Exposition of the Biography of Shākya Chokden (Shākya mchog ldan gyi rnam thar zhib mo rnam par 'byed pa*; hereafter SR): This text devoted to Shākya Chokden is composed by Kunga Drolchok (Kun dga' grol mchog; 1507–1566), who was trained under Shākya Chokden's disciple Donyo Druppa (Don yod grub pa; ca. fifteenth century). The chronicling narration details the cult of this eminent master as well as exhibits his life stories in a comprehensive manner, including but not limited to his traveling, teaching, and interaction with his contemporaries. SR displays detailed information about the full female ordination he conducted.

*The History: How the Teachings were Established in the Three Monasteries at Upper and Lower Tsele (Rtse le gong 'og grwa tshang dgon gsum po rnams kyi bstan pa ji ltar btsugs pa'i lo rgyus*; hereafter RBR): this text is a Buddhist temple gazette, composed by Tsele Natsok Rangdröl (Rtse le sna tshogs rang grol; b. 1608) and records the monastics who made contributions to Tsele monastery. It suggests that a nun named Konchog Tsomo, received full ordination from a male master and imparted teachings to hundreds of nuns, and was revered in Dag (Dwags) valley (Martin 2005, p. 73).

## 3. Narrative Models in Hagiographical Context Concerning Full Female Ordination

The six texts listed above share similarities in terms of their narrative models. When mentioning fully ordained nuns in hagiographies, their male masters, usually being representative in their own lineages, were always serving as essential settings of the narration. They initiated the full ordination and served as the main preceptors (*mkhan po*) of the rituals. The female practitioners, on the other hand, were either distinguished in their schools or their names passed over in silence. In MRZ, CR, SR, and RBR, the male masters only ordained one particular female practitioner, respectively, and their ordinations are recorded. In SGM, MR, and MRR, fully ordained nuns appear in the form of groups, and authors of these texts did not spill much ink on them. To figure out the purposes of composing these writings concerning fully ordained nuns but in different narrative models, I will examine the relevant paragraphs of these texts.

| Case(s) | Relevant Male Master(s) | Female Practitioner(s) |
|---|---|---|
| 1st | Khyungtsang Repa (1115–1176) | Machik Ongjo (ca. 12th) |
| 2nd | Chogle Namgyal (1376–1451) | Chokyi Dronma (1422–1455) |
| 3rd | Mikyod Zhab (u.d.) | Konchog Tsomo (u.d.) |
| 4th | Shākya Chokden (1428–1507) | Choedup Palmo Tso (u.d.) |
| 5th | Rigpa Senge (1287–1375) | A group of fully ordained nuns |
| 6th | Drogon Chogyal Phagpa (1235–1280) | A group of nuns including fully ordained ones |

The first example that comes to us is Machik Ongjo[5]. MRZ records that she was accepted as a lay disciple by Khyungtsang Repa and was recognized as a reincarnation of one of Tilopa's (988–1069) ḍākinī (Tib. mkha' 'gro ma; Martin 2005, p. 67)[6]. Relevant parts in MRZ depict how Machik Ongjo received her full ordination[7]. She appeared as a fully ordained nun when receiving a secret ear-whispered teaching, oral transmission of Rechungpa, for the second time (Rin chen rgya mtsho 1983, p. 286; Martin 2005, p. 67). The text reads:

> At first, [Machik Ongjo] will be a lay disciple and later she will become a fully ordained nun. Because she is a right vessel, [I] shall bestow her [the *gelongma* vow][8].

The narration sketches Machik Ongjo's transformed religious identities, from a lay disciple with a celebrated secular background to a promising fully ordained nun. Though the text narrates a prophecy about her full ordination, the emphasis of the narrative is on her possession of good qualities as a practitioner and her qualifications to receive higher ordination. Otherwise, the narration about her (promising) ordination is lacking in detail and androcentric. It does not provide a further description of her ordination and how she acted as a fully ordained nun in her later life[9]. The authority is given to Khyungtsang Repa to determine whether to confer the ordination.

The following case is set in the fifteenth century. Chokyi Dronma was the princess of Mangyul Gungthang (Mang yul gung thang)[10] a daughter of King Tri Lhawang Gyaltsen (Khri lha dbang rgyal mtshan; 1404–1464). CR alleges that she was recognized as the incarnation of the deity Vajravārāhī by her masters Chogle Namgyal and Thangtong Gyalpo (Diemberger 2007, p. 338). In the related paragraphs of CR, it suggests that Chokyi Dronma was first a semi-ordained nun and subsequently fully ordained (Diemberger 2007, p. 183). In the ceremony, Chogle Namgyal was the principal officiator (Skt. upādhyāya, Tib. *mkhan po*) and Chökyi Wangchuk (Chos kyi dbang phyug; u.d.) acted as the master of the ceremony (Tib. *las kyi slob dpon*). The text reads:

> In the presence of a monastic assembly of faith who are in the required number and endowed with the [right] qualifications, having become a fully ordained nun substantially, her vase of mind was filled with the precepts of excellent disciplines[11].

Similar to Machik Ongjo's family background, Chokyi Dronma was also from a well-known family and then became a distinguished disciple of authoritative masters. CR, in contrast, provides a more detailed description on her full ordination, and it also refers to her as a fully ordained nun in the later parts of the text with that status serving as an honoured appellation (Diemberger 2007). Such a narrative, however, merely tells she was ordained by an assembly of qualified monastics whose gender were uncertain. Moreover, it highlights the male master's positions in the ceremony as it was led by Chogle Namgyal and another monk Chokyi Wangchuk.

Next example is Konchog Tsomo. RBR records that Konchog Tsomo received full ordination from Mikyod Zhab, and she practiced the monastic disciplines perfectly. In Zhongka (Zhong kha) convent, she imparted the Buddhist teachings to more than a hundred ordained women (Tib. *btsun ma*). Because of her distinguished spiritual practice, she was venerated by the people of Dag valley (Martin 2005, pp. 72–73).

The most detailed example among the six cases is provided by SR in terms of depicting who participated in this ordination and their positions. It records that in 1490 when Shākya Chokden went to Gyama (Rgya ma), he and nine other monks conducted the full ordination of Choedup Palmo Tso (Chos grub dpal mo 'tsho; [Kun dga' grol mchog 1974](#), pp. 164–65):

> Shākya Chokden was the principal officiator. The master of the ceremony was acted by Chennga Drupgyal (Sbyan snga grub rgyal). Jetsun Kunga Gyeltsen (Rje btsun kun dga' rgyal mtshan) acted as the mentor (Tib. *gsang ston*). Je Drak Marwa (Rje brag dmar ba) was the master who bestowed the vow of pure conduct (Tib. *tshangs spyod la nyer gnas kyi sdom pa*). Dungwang Zangba (Drung dbang bzang pa) was the one who managed the time of ceremony (Tib. *dus sgo ba drung*), Choeje Samten (Chos rje bsam gtan pa) was the assistant (Tib. *grogs dan pa*), Dungwang Zangba Choeje Samten was the substitute for the master of the ceremony (Tib. *las grwa'i kha skong byes kyi slob dpon*). The four masters conducted the ordination, and Choedup Palmo Tso received the vows of a fully ordained nun[12].

One of the reasons that SR records detailed information of this ceremony might be that the length and content of the Tibetan hagiographical writings had gradually increased over the centuries ([Sernesi 2015](#), p. 738). Otherwise, the narration of the ceremony is presented in an androcentric way, without mentioning any participation of the female *saṃgha*, which is required according to the Mūlasarvāstivāda *Vinaya* tradition. Shākya Chokden later supported the validity of this ordination with his own interpretation of *Vinaya*[13]. Again, this act features the male masters' absolute authority over the ordination and the interpretation of the *Vinaya* tradition.

In the last two cases, fully ordained nuns appear as a group. MRR indicates Rigpa Senge had a group of disciples, who were fully ordained nuns in Minyak of the Kham area. The text reads:

> At that time, Darma Gyeltsen (Dar ma rgyal mtshan), Gyalba Pel (Rgyal ba dpal), Shākya Pel (Shākya dpal), Wangyal (Byang rgyal), Kunga Gyeltsen (Kun dga' rgyal mtshan), Rgyal (Bla rgyal), Dawa (Zla ba) and so forth were there. Having shaved, the fully ordained nun Tashi Pel (Bkar shi dpal), as a [female] disciple, appeared at Rabgang (Rab sgang) [monastery], [and] because of this reason, a monastic [community] of fully ordained nuns was also nurtured[14].

Though the text does not assert Tashi Pel participating in the full ordination of nuns, she might have had the role of educating a new female monastic assembly (Tib. *dge slong ma'i dge 'dun*). Later parts of the text suggest that Rigpa Senge was the teacher of an assembly of fully ordained nuns, who studied "A Guide to the Bodhisattva's Way of Life" (Bodhicaryāvatāra), observed monastic discipline, and did meditation. Because of the merits of practice as such, he was able to live longer ([Seng ge bzang po 1983](#), p. 61; [Price-Wallace 2015](#), pp. 230–31)[15]. These two paragraphs depicting the activities of fully ordained nuns serve mostly to highlight the spiritual achievements and merits of Rigpa Senge.

The last case is recorded in SGM. One paragraph in this text praises Drogon Chogyal Phagpa's (1235–1280) achievement in *Vinaya* practice by mentioning that he ordained 1425 monastics, including fully ordained monks and nuns, semi-ordained monks and nuns ([Chen 2005](#), p. 150). However, details of the full ordinations of nuns given by Drogon Chogyal Phagpa remain unknown.

Paragraphs extracted from SGM, MRR, and SR feature the achievement of eminent male masters. These narratives mention the existence of fully ordained nuns mainly to glorify their male masters' outstanding spiritual performance.

By examining the three hagiographies featuring female practitioners, namely MRZ, CR, and RBR, I found out that though the writings center on these female personas, and they are alleged to have been fully ordained, the relevant narration about their ordinations is obscure, and their male masters still end up at the center in terms of the instrumental roles they play in their ordinations. Besides, the focus of the narrative is on female practitioners'

possession of good qualities, their qualification to receive higher ordination, and their excellent spiritual achievement afterward. These phenomena imply that full ordinations might have been carried out according to their masters' expectations, and fully ordained nuns facilitate their revered status in their own schools. Such monastic identity is honorific rather than practical because they were acting as leaders of their female peers enjoyed the rare access to full ordination. The methods of their full ordinations might not strictly follow the Mūlasarvāstivāda *Vinaya* tradition as this ceremony should be conducted by both male and female monastic assemblies (Tsedroen and Anālayo 2013, p. 761). We could thus infer that these women being able to access to full ordination highlight their distinction in spiritual performance. The purpose of composing the narrations of them being fully ordained nuns is to establish the religious status in their schools. Meanwhile, the religious positions of female disciples are always highlighted by their male masters who possessed the power and authority of schools, and these women inherit parts of the authority because like Machik Ongjo and Chokyi Dronma, they did not only have rare access to full ordination but also became important figures in their respective schools as Machik Ongjo was one of the lineage holders of the oral transmission of Rechungpa and Chokyi Dronma is recognized as the first Samding Dorje Pakmo[16]. When fully ordained nuns appear in hagiographies devoted to male masters, such as in SGM, MR, and SR, relevant descriptions of these nuns serve to underline the masters' greatness in spiritual practice, whereas usually, these females were less famous, and most of their names are passed over in silence.

Additionally, we could observe that from the examples of Machik Ongjo to Choedup Palmo Tso, the descriptions of their ordinations are more detailed, developing from a mere brief sentence to a well-documented narrative displaying who got involved and what their functions were in the ceremony. When it comes to the crucial points, such as whether other fully ordained nuns participated in the ordinations, however, all the authors keep silent. Such silence could have two interpretations. First, these biographies highlight male masters dominating those full female ordinations in order to feature those masters' religious authority. Therefore, who conferred ordinations weighs more than how ordinations were conducted. The other possibility is that the Mūlasarvāstivāda *Vinaya* tradition requires that the ceremonies (Skt. *karmavastu*, Tib. *las kyi gzhi*), involving more than four people, should be conducted by both male and female monastic assemblies (Shi 1999b, p. 359). However, it is generally believed that the lineage of fully ordained nuns was never introduced to Tibet, indicating there might not have ever been fully ordained nuns who could participate in such rituals. Consequently, the authors of these hagiographies might consciously use ambivalent nouns such as monastic assemblies (Tib. *dge 'dun*), which could refer to both male and female monastic assemblies in order to make such narrations less controversial[17].

## 4. Conclusions

Full female ordination has been unavailable in the Tibetan Buddhist community, and thus the nuns following this tradition are restricted to semi-ordination. Recently, however, Je Khenpo (b. 1966), the spiritual leader of Bhutan bestowed full female ordination to 144 nuns. Whether this practice could (re-)introduce the order of fully ordained nuns to Tibetan Buddhism for good is still to be seen[18]. For this issue, prior scholarship has located a few fourteenth-to-seventeenth-century hagiographical writings. They suggest the existence of fully ordained Tibetan nuns, who were alleged to live in the twelfth to sixteenth centuries. The recognition of monastic identities as such, however, is disputed because the method of their ordinations, probably conferred by the assembly of fully ordained monks alone, does not strictly follow the Mūlasarvāstivāda *Vinaya* tradition, which is practiced by the Tibetan Buddhists. In an effort to learn why these female practitioners are depicted as fully ordained nuns, I revisited relevant hagiographies with particular reference to *The Biography of Chokyi Dronma* and *A Detailed Exposition of the Biography of Shākya Chokden*. Based on the findings, this paper suggests that the hagiographies, which feature outstanding female practitioners, depict their subjects as fully ordained nuns to

serve the purpose of highlighting their outstanding spiritual achievement. On the other hand, in order to underline male masters' distinguished religious practices, hagiographies devoted to them portray the image of fully ordained nuns. The narrations of these nuns, in turn, are subordinate, and thus, these females' full ordinations are briefly depicted. Besides, the hagiographical writings employ ambivalent expressions to portray the detail of full female ordinations, such as using gender-neutral expressions to refer to who conducted the rituals, most probably for the purpose of making the narration less controversial doctrinally.

In a nutshell, these writings are androcentric by nature and emphasize the eminence of relevant male masters. Although they depict a series of fully ordained nuns, the acceptance of these females as being gelongmas may not go beyond the context of these hagiographies[19]. For those of us interested in how Tibetan hagiographies of male figures influence the literary history of its women, this paper may serve as an example of work and such research can happen on a larger scale through current digital means such as Buddhist Digital Resource Center[20].

**Funding:** This research received no external funding.

**Institutional Review Board Statement:** Not applicable.

**Informed Consent Statement:** Not applicable.

**Data Availability Statement:** All primary sources in Tibetan are digitally persevered in The Buddhist Digital Resource Center, https://www.bdrc.io, accessed on 20 September 2022.

**Acknowledgments:** I would first express my sincere gratitude to David Germano and Kurtis Schaeffer for their insightful feedback and gracious support of my paper. I am grateful to anonymous reviewers for their valuable comments. I would extend my special thanks to Phuntsok Wangchuk and Shiu Long In for their suggestions on earlier drafts.

**Conflicts of Interest:** The author declares no conflict of interest.

## Notes

1　　High-ranking Tibetan masters encourage their nuns to receive higher ordination from Chinese nuns, who preserve the tradition of such ordination. Besides, Sakyadhita, a large-scale international group of Buddhist nuns, organized conferences on this issue. They were held in Thailand, Sri Lanka, India, Cambodia, Nepal, Taiwan, Malaysia, Mongolia, and Vietnam from 1991 to 2010. See (Schneider 2012). Extant scholarship explores possible solutions to (re-)introduce this lineage to Tibetan Buddhism. One suggestion is to bestow the full ordination by Tibetan monks alone. See (Shi 1999a; Tsedroen and Anālayo 2013; Bodhi 2010; Tsering 2010; Ryōji 2015). Another suggestion is to invite Chinese fully ordained nuns to co-conduct the ordination with Tibetan monks. See (Shi 1999a; Chodron 2010; Sujato 2010).

2　　The reason I use (re-)introduce here is because Tibetan Buddhist masters might have conferred full female ordination that was shown in their hagiographies. If they did confer such ceremonies, regardless such practice strictly followed the Mūlasarvāstivāda *Vinaya* tradition or not, the contemporary effort to establish the order of fully ordained nuns in Tibetan Buddhist community is to reintroduce the full female ordination. Extant scholarship, however, indicates that the order of fully ordained nuns may not have been firmly established in Tibet (Havnevik 1989, p. 45; Skilling 1994, p. 36; Campbell 1996, p. 5). In that case, establishing such a monastic order is to introduce this ritual practice to Tibetan Buddhism.

3　　Among the existing textual evidence, examples of fully ordained nuns are mainly found in scriptural texts which date from the fourteenth to seventeenth centuries. More hagiographies concerning fully ordained nuns might exist, but these exceed the bounds of this paper as this essay specifically focus on the fourteenth-to-seventeenth-century textual sources.

4　　Rigpa Senge was one of the five learned scholars of Minyak. The other four are Mase Tönpa (Rma se ston pa; 1317–1383), Gyalwa Rinchen (Rgyal bar in chen; 1328–1386), Chukmo Tönpa (Phyug mo ston pa; 1332–1392), and Jamsar (Byam gsar; 1318–1386).

5　　She was born into a wealthy family allegedly belonging to the Gyamo (Rgya mo) clan in Uyuk ('U yug), who owned a great deal of land, livestock, and crops (Allione 2000, p. 296; Martin 2005, p. 66). From early life, although Machik Ongjo held a pessimistic view towards the mundane life, she had great faith in Buddhism. Despite that Machik Ongjo got married, she was tormented by household life and finally renounced the world. See Martin (2005, p. 67).

6　　Dākinī is sometimes used interchangeably with the term "yoginī", particularly in the tantric Buddhist context. It means "goddesses with magical abilities". See Nobumi and Jansen (2019, p. 132).

7　　Five Tibetan sources surrounding Machik Ongjo exist till date. Among which, four are biographies: MRZ (Anon 1983, pp. 285–88), *The Biography of Machik Ongjo* (Ma gcig ong jo'i rnam thar; MRT; Mkhas btsun bzang po 1973, p. 52), *The Blue Annals* (Roerich 1953,

pp. 443–46) and *Khyngtsangpa's Disciple Machik Ongjo* (Khyung tshang pa'i slob ma ma gcig ong jo; KSM; 'Phrin las rgya mtsho 2009, pp. 359–60). One source, *Tibetan chronicle of Pema Karpo* (*Chos 'byung bstan pa'i pad+ma rgyas pa'i nyin byed*, compiled by Pema Karpo (Padma dkar po) in 1592, just mentions her in a single sentence (Padma dkar po 1968, p. 509). MRZ is probably the earliest one among the four biographies, albeit its compiler and dating remain unknown. It is the most extensive source among the others about Machik Ongjo by providing significant information about her spiritual life. The second source appears in *The Blue Annals*, which was compiled in 1476 by Go Zhonnu Pel (Gzhon nu dpal; 1392–1481). The third one MRT, was written by Trinle Gyatso ('Phrin las rgya mtsho; u.d.) in 1845 and the last Tibetan source, KSM, was composed by a contemporary scholar Khetsun Zangpo (Mkhas btsun bzang po; b. 1920).

[8]   de yang dang po dge bsnyen phyis dge slong du yong/de snod ldan yin pas byin gsungs so/See Rin chen rgya mtsho (1983, p. 286, l. 5).

[9]   It is also worthy to note that in a hagiographical reference devoted to Padmasambhava (ca. 8th) depicts his two tantric consorts, Yeshe Tsogyel (Ye shes mtsho rgyal; ca. 8th) and Mandarava (ca. 8th) as fully ordained nuns. A tantric consort, however, is supposed to participate in sexual unions, so it might not be possible for her to be a fully ordained nun who would live a celibate life. Liang suggests that the narrative aims at underlining Yeshe Tsogyel and Mandarava's purity that they are free from secular defilements, it therefore, portrays them as fully ordained nuns who possess more merits and independence. See Liang (2020, p. 221). Similar to Machik Ongjo's case, the author of MRZ might also intend to highlight Machik Ongjo's religious purity, just like a fully ordained nun.

[10]  In southwest Tibet.

[11]  dad pa'i dge 'dun grangs dang mtshan nyid yongs su rdzogs pa'i dbus su tshigs phyi ma dge slong ma'i dngos por bsgrubs nas lhag pa tshul khrims kyi bslab pas thugs kyi bum pa gang ste/ See (Anon 2018, f.60b, line 2–3).

[12]  pan chen rin po che'i drung du mkhan po zhus/las kyi slob dpon sbyan snga grub pa'i rgyal po/gsang ston rje btsun kun dga' rgyal mtshan/tshangs spyod nyer gnas kyi slob dpon rje brag dmar ba/ dus sgo ba drung dbang bzang pa/grogs dan pa chos rje bsam gtan pa/las grwa'i kha skong byes kyi slob dpon pa rnam bzhis mdzad nas/dge slong ma'i sdom pa bzhes/See Kun dga' grol mchog (1974, p. 164, l. 5–8).

[13]  Shākya Chokden validated such ordination because the full ordination of women has two parts: the first is conferred by an assembly of fully ordained nuns, and the second is conducted by an assembly of fully ordained monks, and thus they can be interpreted separately. He further argued that the prerequisites of being a probationary nun (Skt. *śikṣamāṇā*, Tib. *dge slob ma*) and receiving the vows of chastity are necessary when the candidates are ordained by assemblies of both fully ordained monks and fully ordained nuns, but that such practices are not compulsory for ordination by an assembly of fully ordained monks alone. See https://thubtenchodron.org/2006/05/mulasarvastivada-bhikshuni/#rf1-54152 (accessed on 26 July 2022).

[14]  de dus dar ma rgyal mtshan/rgyal ba dpal/ shākya dpal/ byang rgyal/ kun dga' rgyal mtshan/bla rgyal/zla ba sogs byung zhing spu 'bor nas gdul bya'i yan lag tu dge slong ma bkra shis dpal rab sgang du byung ba la brten nas dge slong ma'i dge 'dun yang bskyangs so/ See Seng ge bzang po (1983, p. 36, l. 4–5).

[15]  dge slong ma'i dge 'dun brgya phrag mang pos spyod 'jug dang /'dul pa la sogs kyi 'chad nyan dang /sgom sgrub la rtse cig tu gzhol bas sgrub pa'i mchod pas mnyes pa sgrub par byed kyin yod pa/ See Seng ge bzang po (1983, p. 61, l. 5–6).

[16]  Chokyi Dronma's reincarnation Kunga Zangmo (Kun dga' bzang mo; 1459–1502) initiated an incarnation lineage of the Samding Dorje Pakmo.

[17]  For male ordination, normally a semi-ordained monk (Tib. *dge tshul*) needs to receive the vows from ten fully ordained monks who have been ordained for at least ten years. In remote areas, however, the minimum number of required monks can be reduced to five. Lachen Gongpa Rabsal (Bla chen dgongs pa rab gsal; 825–915), for instance, was ordained by five monks. See Thubten Chodron (2010, pp. 185, 191–92).

[18]  For more information about this ordination, see https://www.lionsroar.com/women-receive-full-ordination-in-bhutan-for-first-time-in-modern-history/ (accessed on 22 July 2022). Additionally, see a document published by Je Khenpo https://www.facebook.com/bhutantimes1/posts/1904123783106794 (accessed on 21 September 2022).

[19]  Machik Ongjo's case can support this view because only one hagiography devoted to her mentions that she had received the full ordination, while the other four sources keep silent on this point.

[20]  The Buddhist Digital Resource Center is a nonprofit organization which aims at preserving and disseminating Buddhist literature to scholars as well as practitioners. Most of their online manuscripts are in Tibetan language. https://www.bdrc.io, accessed on 20 September 2022.

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
