# Peer review of "Fully Ordained Nuns in Fourteenth-to-Seventeenth Tibetan Hagiographical Narratives"

_religions, doi:10.3390/rel13111037_

Round 1
Reviewer 1 Report
This article provides an original contribution to the study of women's Buddhism by examining the ordination narratives of a series of fully ordained Tibetan Buddhist nuns. Instead of accepting fully ordained status as a static condition, the author critically examines the circumstances enabling the full ordination and the intended literary and social impact of the ordination. In addition, the author demonstrates their linguistic abilities and a good command of the primary sources.
Two comments that the author might want to consider to strengthen their argument:
1. In the life stories of Yeshe Tsogyel and Mandarava, they are both referred to as fully ordained nuns (dge slong ma). Since both women are consorts to Padmasambhava, it is unlikely for them to remain in a celibate status.
For example, O rgyan gling pa's Padma bka’ thang (Chengdu: Si khron mi rigs dpe skrun khang, 1987), on p.705: Yeshe Tsogyel claimed herself to be a Gelongma (dge slong ma yin ’khor ba’i skyon ma gos); on p.251 of the same text, Mandarava is also depicted as a fully ordained nun. This identity of consorts as fully ordained nuns is discussed in Liang 2020, 214-217.
It might be worthwhile to consider how to use this evidence to support the author's argument of full ordination as an honorific status.
2. On p.7 of the reviewer copy, the last paragraph before "4. Conclusions." The reviewer wonder if the author would also like to refer to the ordination record of male masters and make a brief comparison to strengthen the two hypotheses in this paragraph.
Lastly, the article will also benefit from fixing some inconsistencies in Sanskrit diacritics, Tibetan phonetic renderings, and Tibetan transliterations.
1. Sanskrit: the diacritics for the Sanskrit word ḍākinī is added inconsistently or not added at all, so is yoginī. And if diacritics are adopted, it should be shākya instead of shAkya.
2. Tibetan phonetics: p.4, in the table, if ö is used for Chökyi Drönma, shouldn't it also be Chödup Palmo Tso instead of Choedup Palmo Tso? Same for p.5 line 196, Choeje Samten.
3. Tibetan transliteration: the smart quotes and straight quotes are used inconsistently.
A few minor typos:
p.1 line 30 Price-Wallace this volume => Price-Wallace in this volume
p.3 line 91 the Sakya school => the Sakya School
p.4n7, line 2 Ma gcig ang jo => Ma gcig ong jo
Author Response
Thank you for your comments!
For the first point, I have added Yeshe Tsogyel's example in footnote 9 as my article mainly focuses on the hagiographies, dating from the 14th to 17th century, featuring Tibetan women practitioners.
For the second point, I have added Gongpa Rabsal's case as an example of male ordination
I have corrected the inconsistence and typos you mentioned, thank you for pointing them out!
Reviewer 2 Report
This is a very relevant paper, particularly considering the recent full ordination ceremony that took place in Bhutan in the summer of 2022. The author does mention this ceremony at the end of the article, but it might be worth mentioning it at the beginning too (or even in the abstract).
The abstract might consider framing the main argument of the article in the context of the contemporary debate about the existence (or not) of the full ordination lineage for nuns in the Himalayan region.
There are some editing issues that need to be solved. Footnote 2 starts in a strange way. In line 40 the author uses the term "rendered" when I think it should be "translated." Footnote 3 also could use some editing. Line 244 has a lingering "(?)" that needs to be solved. These are just some examples.
I would argue that the most valuable contribution of the article is the unearthing of the six references to full ordination ceremonies in 14th to 17th-century sources. Tha author does a good job analyzing those references and establishing that the process and impact of those so-called ordinations might have been more honorific than practical, but they are still relevant in the context of the current conversation about precedent when it comes to full ordination of nuns today.
The table with cases, master, and fully ordained nuns could include dates (and maybe lineage?).
The author mentions that a discussion as to why the sources chosen are from the 14-17 centuries is "a moot point." I think it is not moot and the author should either clarify or simply mention that these are the sources the author has found and more might exist (which the author acknowledges).
The article is quite short, and as a reader, I am left with a feeling that much more could have been done with the primary sources.
Not sure why the author uses "semi-ordination" instead of "novice ordination."
The issue of introducing vs. re-introducing full ordination is an interesting one, but the author should phrase this more clearly.
The conclusion talks about the recent full ordination that took place in Bhutan. The author might want to look at the document published by the Je Khenpo to justify the ceremony. It includes references to some of the ordinations mentioned in this article.
Overall, though, I think this is a great contribution to our current understanding of the issue of full ordination in the Tibetan Buddhist tradition." Also very relevant.
Author Response
Thank you for your comments!
In the revised version of the abstract, I will mention the issue of the existence (or not) of the full ordination lineage for nuns in the Himalayan region.
Regarding the second comment, I will correct them.
Regarding the dates or lineage of female practitioners, I will mention them in the main body of the text, but in the revised version I have added the dating as well.
In the revised version I will clarify the moot point.
I will adopt novice ordination in my revised version.
I will clarify the issue of introducing vs. re-introducing full ordination.
Thank you for reminding the document published by Je Khenpo, he does mention some ordinations I examine in this article, but I think he uses these examples as textual support.
Reviewer 3 Report
excellent work, great contribution
Author Response
Thank you for your kind comment!